# Factors Influencing Early Marginal Bone Loss around Dental Implants Positioned Subcrestally: A Multicenter Prospective Clinical Study

**DOI:** 10.3390/jcm8081168

**Published:** 2019-08-04

**Authors:** Teresa Lombardi, Federico Berton, Stefano Salgarello, Erika Barbalonga, Antonio Rapani, Francesca Piovesana, Caterina Gregorio, Giulia Barbati, Roberto Di Lenarda, Claudio Stacchi

**Affiliations:** 1Private Practice, 87011 Cassano allo Ionio, Italy; 2Department of Medical, Surgical and Health Sciences, University of Trieste, 34129 Trieste, Italy; 3Department of Medical and Surgical Specialties, Radiological Sciences and Public Health, University of Brescia, 25123 Brescia, Italy; 4Private Practice, 6600 Locarno, Switzerland; 5Department of Statistics, University of Padova, 35121 Padova, Italy

**Keywords:** abutment height, subcrestal implants, marginal bone loss, implant insertion depth, vertical mucosal thickness, biological width

## Abstract

Early marginal bone loss (MBL) is a non-infective remodeling process of variable entity occurring within the first year after implant placement. It has a multifactorial etiology, being influenced by both surgical and prosthetic factors. Their impact remains a matter of debate, and controversial information is available, particularly regarding implants placed subcrestally. The present multicenter prospective clinical study aimed to correlate marginal bone loss around platform-switched implants with conical connection inserted subcrestally to general and local factors. Fifty-five patients were enrolled according to strict inclusion/exclusion criteria by four clinical centers. Single or multiple implants (AnyRidge, MegaGen, South Korea) were inserted in the posterior mandible with a one-stage protocol. Impressions were taken after two months of healing (T1), screwed metal-ceramic restorations were delivered three months after implant insertion (T2), and patients were recalled after six months (T3) and twelve months (T4) of prosthetic loading. Periapical radiographs were acquired at each time point. Bone levels were measured at each time point on both mesial and distal aspects of implants. Linear mixed models were fitted to the data to identify predictors associated with MBL. Fifty patients (25 male, 25 female; mean age 58.0 ± 12.8) with a total of 83 implants were included in the final analysis. The mean subcrestal position of the implant shoulder at baseline was 1.24 ± 0.57 mm, while at T4, it was 0.46 ± 0.59 mm under the bone level. Early marginal bone remodeling was significantly influenced by implant insertion depth and factors related to biological width establishment (vertical mucosal thickness, healing, and prosthetic abutment height). Deep implant insertion, thin peri-implant mucosa, and short abutments were associated with greater marginal bone loss up to six months after prosthetic loading. Peri-implant bone levels tended to stabilize after this time, and no further marginal bone resorption was recorded at twelve months after implant loading.

## 1. Introduction

A complex cascade of biological events occurs after implant insertion. In this type of surgery, wound healing response after surgical trauma is conditioned by the presence of foreign material in the host bone. According to studies by Donath [1,2], a foreign material inside the human body may elicit four types of host response: rejection, dissolution, resorption, or demarcation. Demarcation, which represents a protective reaction aiming to separate a foreign body impossible to dissolute or resorb from healthy tissue, usually results in fibrous encapsulation. However, when biocompatible material is surrounded by bone in a protected environment (with neither infection nor micromovements), bone encapsulation usually occurs, forming a robust bone-to-implant interface, which can be used for clinical purposes: the osseointegration phenomenon [3]. The majority of osseointegrated implants show successful long-term clinical outcomes due to the establishment of steady-state bone remodeling activity. However, the condition of foreign-body equilibrium may be compromised by various factors at different times. The main clinical sign of imbalance between bone apposition and resorption is a marginal bone loss (MBL) [4]. Marginal bone stability around dental implants has always been considered one of the main criteria for defining implant success [5].

Early MBL is a non-infective remodeling process of variable entity occurring within the first year after implant placement. It has a multifactorial etiology, being influenced by both surgical factors (insufficient crestal width and/or implant malpositioning, bone overheating during implant site preparation, implant crest module characteristics, excessive cortical compression) and prosthetic variables (type of implant/abutment connection, entity and location of implant/abutment microgap, number of abutment disconnections, abutment height, residual cement, early loading) [6,7,8,9]. Early MBL represents an adaptive response of peri-implant marginal bone to the combined effect of these factors and has been considered to have an important prognostic value for predicting long-term implant success. A recent study on implants positioned at the crestal level suggested that early MBL >0.44 mm in the first six months after prosthetic loading is a risk indicator for peri-implant bone loss progression [10].

Modifications of horizontal and vertical relationships between implant-abutment junction (IAJ) and bone crest have been suggested to influence the entity of early MBL. The horizontal displacement of the microgap location far from the bone crest using an abutment narrower than the implant neck (platform switching) has been demonstrated to be effective in reducing early MBL [11,12]. Conversely, controversial information is available regarding implants placed subcrestally. Some authors recommended placement of the implant platform 1 or 2 mm below the alveolar crest to better maintain marginal bone levels [13,14]. However, other studies reported an increased extension of inflammatory infiltrate due to deep positioning of the IAJ, resulting in greater MBL compared to implants placed equicrestally [15,16].

Therefore, the primary aim of the present multicenter prospective clinical study was to analyze factors potentially influencing early MBL around platform-switched implants with conical connection inserted subcrestally, up to 15 months after implant placement.

## 2. Material and Methods

### 2.1. Study Protocol

This multicenter prospective clinical study was reported in strict adherence to the criteria of the STROBE (Strengthening the Reporting of Observational Studies in Epidemiology) checklist. All procedures were performed per the recommendations of the Declaration of Helsinki, as revised in Fortaleza (2013), for investigations with human subjects. The study protocol was approved by the relevant Ethical Committee (Regione Calabria, Sezione Area Nord, No. 46/2016), and was recorded in a public register of clinical trials (www.clinicaltrials.gov-NCT03077880). All eligible patients were thoroughly informed of the study protocol (including surgical and prosthetic procedures, follow-up visits, potential risks involved, and possible therapeutic alternatives), and signed an informed consent form. Patients authorized the use of their data for research purposes.

### 2.2. Selection Criteria

Any partially edentulous patient requiring implant therapy for fixed prosthetic rehabilitation in the posterior mandible was eligible for this study, subject to the following inclusion and exclusion criteria.

General inclusion criteria were: (I) age >18 years; (II) good general health; (III) patient willing and fully capable to comply with the study protocol; (IV) written informed consent given.

Local inclusion criteria were: (I) presence of keratinized mucosa with a minimum buccolingual width of 4 mm; (II) bone crest with at least 6 mm width and 8 mm height above the mandibular canal in the site when the implant was planned; (III) healed bone crest (at least 6 months elapsed from tooth extraction); (IV) no grafted bone; (V) full mouth plaque score (FMPS) <25% and full mouth bleeding score (FMBS) <20%; (VI) implant insertion torque (IT) >20 Ncm; (VII) presence of the opposing dentition; (VIII) subcrestal positioning. Exclusion criteria were: (I) history of head or neck radiation therapy; (II) uncontrolled diabetes (hemoglobinA1c >7.5%); (III) immunocompromised patients (HIV infection or chemotherapy within the past 5 years); (IV) present or past treatment with intravenous bisphosphonates; (V) patient pregnancy or lactating at any time during the study; (VI) psychological or psychiatric problems; (VII) alcohol or drugs abuse; (VIII) participating in other studies, if the present protocol could not be properly followed.

All patients, selected consecutively between April 2016 and October 2017, were treated independently by four operators (T.L., S.S., E.B., and C.S.) in four private offices. Data collection was performed by a single independent examiner (F.B.).

All patients received oral hygiene instructions and underwent deplaquing 1 week before surgery. Cone beam computed tomography (CBCT) was performed to analyze available bone volume and to plan implant insertion.

### 2.3. Surgical and Restorative Procedures

All patients were administered with antibiotic prophylaxis (amoxicillin 2 g) one hour before surgery. A mid-crestal incision was performed under local anesthesia, preserving an adequate quantity of keratinized tissue on both buccal and lingual sides. A full-thickness buccal flap was elevated, and vertical mucosal thickness of the undetached lingual flap was measured with a periodontal probe at the center of the programmed implant site, as described elsewhere [17]. The lingual flap was subsequently elevated, and implant site preparation was performed under abundant irrigation of cold saline solution following the manufacturer’s recommendations for subcrestal placement. Platform-switched implants with conical connection (AnyRidge, MegaGen, Gyeongbuk, South Korea) were inserted under bone level, and peak IT values were recorded by the surgical motor (Implantmed, W&H, Burmoos, Austria). Implants were immediately connected to healing or transepithelial abutments (Octa, MegaGen, Gyeongbuk, South Korea), adapting their length to the site-specific soft tissue vertical thickness. Flaps were sutured with single stitches and Sentineri technique using synthetic monofilament [18]. Patients were prescribed post-surgical antibiotic therapy (amoxicillin 1 g twice a day) for six days, and nonsteroidal anti-inflammatory drugs (ibuprofen 600 mg), when necessary. Sutures were removed 10–14 days after surgery. Patients were instructed not to use removable prostheses during the entire healing period.

After two months of healing, implants were clinically and radiographically evaluated, and final impressions were taken. Prosthetic abutments height was chosen, adapting their length to the site-specific soft tissue vertical thickness. After functional and aesthetic try-in, screw-retained metal-ceramic rehabilitations were delivered.

Periodontal status was assessed using the modified plaque index (mPI) and modified sulcus bleeding index (mSBI) at prosthesis delivery and after 6 and 12 months of functional loading [19]. The mean of the four values recorded for each implant (mesial, distal, buccal, and lingual) was subsequently analyzed.

### 2.4. Radiographic Measurements

Digital radiographs were taken using a long-cone paralleling technique with a Rinn-type film holder at the time of implant placement (baseline—T0), at impression taking (2 months after implant placement—T1), at prosthetic restoration delivery (3 months after implant placement—T2), and after 6 and 12 months of prosthetic loading (T3 and T4, respectively) (Figure 1).

The distance between IAJ and bone crest was measured at each time interval, on both mesial and distal aspects of the implant. A positive value was assigned when the bone crest was coronal to the IAJ, whereas a negative value was assigned when the bone crest was apical to the IAJ.

Any radiograph showing signs of deformation or poor image quality was immediately repeated. All measurements were taken by a single calibrated examiner (F.P.), on a 30-inch led-backlit color diagnostic display, using measuring software (Image J 1.52a, National Institutes of Health, USA) (Figure 2). Each measurement was repeated three times at three different time points, as proposed by Gomez-Roman and Launer [20]. Examiner calibration was performed by assessing ten radiographs, with a different author (F.B.) serving as a reference examiner. Intra-examiner and inter-examiner concordances were 91.9% and 85.2%, respectively, for linear measurements within ±0.1 mm.

### 2.5. Predictor and Outcome Variables

Primary predictor variables and their respective period of activity were evaluated as follows: (i) vertical mucosal thickness (thick >2 mm vs. thin ≤2 mm; from T0 to T4); (ii) implant insertion torque (>50 Ncm vs. ≤50 Ncm; from T0 to T1); (iii) depth of implant insertion (mm; from T0 to T4); (iv) healing abutment height (long ≥3 mm vs. short <3 mm; from T0 to T2); (v) number of abutment disconnections (zero vs. multiple; from T0 to T4); (vi) prosthetic abutment height (long ≥3 mm vs. short <3 mm; from T2 to T4); (vii) type of prosthetic restoration (single crown vs. short-span bridge; from T2 to T4). The influence of the following patient-related variables, possibly correlated with the predictor and outcome variables, were also evaluated from T0 to T4: (i) age; (ii) gender; (iii) smoking status (smoker vs. no smoker); (iv) periodontal status (periodontal health vs. chronic periodontitis).

Primary outcome (dependent variable):early MBL (up to 12 months from prosthetic loading).

Secondary outcomes:implant failure: implant mobility or implant removal due to progressive marginal bone loss. Implant stability was tested by tightening abutment screws (35 N/cm) at prosthesis delivery.any complication or adverse event.

### 2.6. Statistical Analysis

Statistical analysis was performed using R software version 3.5.3 (nlme package; version 3.1.140). Early MBL at each time point was defined as the difference between the depth of implant insertion at the time and depth of implant insertion at baseline.

The means of mesial and distal early MBL at each follow-up were compared using a *t*-test for paired data. No significant differences were found (T1: Toss = 0.413, d.f. (degrees of freedom) = 82, *p* = 0.68; T2: Toss = 0.002, d.f. = 82, *p* = 1; T3: Toss = –0.143, d.f. = 82, *p* = 0.89; T4: Toss = –0.027, d.f. = 82, *p* = 0.98), hence the mean between mesial and distal MBL was used as a primary outcome in subsequent analysis.

Linear mixed models were fitted to the data to identify predictors associated with MBL. Since not all variables were active at all time-points, four different models were estimated. First, a global model (Model A) was built, considering all four follow-up time points. Preliminary models with only one covariate at a time, plus time and peri-implant bone level at T0, were estimated to select the covariates to be included in the final multivariable model. Only covariates with *p* < 0.05 entered the final model. Covariates taken into account were those that might have affected the primary outcome over the whole follow-up period: age, gender, smoking status, periodontal status, vertical mucosal thickness, and multiple abutment disconnections. Time was considered a continuous variable measured in months, and was modeled using a linear spline function with one knot at T3, as indicated by graphical preliminary exploratory analyses. Subsequently, models considering different follow-up times were estimated:Model B: T1Model C: T1, T2Model D: T2, T3, T4

In these models, in addition to the covariates used in Model A, other independent variables active only at specific time points were added (implant insertion torque in Model B; healing abutment height in Model C; prosthetic abutment height and type of prosthetic restoration in Model D).

In models C and D, time was used as a categorical variable. Time was not included in Model B as only one MBL measurement was involved. Covariate selection was performed using preliminary models, as explained above.

The random-effects structure of each model was evaluated using the Likelihood Ratio Test. For all models, the most suitable structure was random intercept on the subject and random intercept on the operator, except for Model B, in which only random intercept on the operator was included.

## 3. Results

### 3.1. Demographics and Clinical Outcomes

Of a total of 228 patients evaluated for eligibility, 55 consecutive patients (T.L. 20; S.S. 9; E.B. 6; C.S. 20) fulfilled all inclusion/exclusion requirements and were enrolled in the present study. Included patients (27 male, 28 female; mean age 57.3 ± 12.6, age-range 32–85) were treated between June 2016 and March 2017 with the insertion of a total of 91 implants in the posterior mandible. Two implants in two patients were not placed subcrestally (negative value of the mean between mesial and distal measurements). Three implants in three patients failed to osseointegrate and were removed before taking impressions (primary failure rate of 3.3%). Two patients (three implants) were lost at 6-month follow-up (one patient of T.L. was jailed, and one patient of C.S. moved abroad). Finally, fifty patients (25 male, 25 female; mean age 58.0 ± 12.8, age-range 32–85), with a total of 83 implants that received single or short-span screwed metal-ceramic restorations, completed all phases of the study and were included in the final analysis. The final sample was balanced in terms of age and gender distribution. Complete demographics and characteristics of the included patients are summarized in Table 1.

No complications or adverse effects were recorded, and no additional implants were lost. All 83 implants were functioning satisfactorily 6 months and 12 months after prosthetic loading. Although slight increases in mPI and mSBI values were recorded at 15-month follow up, no significant differences were recorded between T3 and T4.

### 3.2. Marginal Bone Level Changes

The mean subcrestal position of the implant shoulder at T0 was 1.24 ± 0.57 mm under bone level, while at T4, it was 0.46 ± 0.59 mm. At the end of the follow-up period, out of a total of 83 implants, the platform of 63 implants remained subcrestal (75.9%), four resulted in equicrestal (4.8%), and 16 resulted in supracrestal (19.3%).

The pattern of marginal bone resorption over time was the following: mean MBL at T1 was 0.5 ± 0.34 mm. An additional mean MBL of 0.18 ± 0.22 mm was registered at T2. A further increase in mean MBL of 0.11 ± 0.20 mm occurred at T3, while complete marginal bone stabilization was observed at T4 (mean MBL compared to T3 = 0.00 ± 0.19 mm). Marginal bone levels at the different time-points are represented in Figure 3.

Results of preliminary linear mixed models analyzing factors possibly influencing MBL from T0 and T4 are reported in Table 2. From T0 to T4, no significant relationships between early MBL and patient age, gender, smoking status, periodontal status, and multiple abutment disconnections were demonstrated. By contrast, implant insertion depth and presence of thin peri-implant mucosa demonstrated significant negative influence upon early MBL from T0 to T4 (*p* < 0.01 and *p* = 0.01, respectively). Complete results of the multivariable analysis are reported in Table 3.

MBL variability within patients and operators was evaluated by a random-effects model and, from T0 to T1, was observed to be 0.25 mm (0.19; 0.34) and 0.21 mm (0.13; 0.35), respectively.

Preliminary linear mixed models analyzing factors possibly influencing MBL within specific time frames showed no significant associations between implant insertion torque and MBL (from T0 to T1; *p* = 0.62) or between type of prosthetic restoration (single crown vs. short-span bridge) and marginal bone levels (from T2 to T4; *p* = 0.92). Conversely, healing abutment height (from T0 to T2) and prosthetic abutment height (from T2 to T4) had a significant influence on early MBL. In particular, short healing and prosthetic abutments (<3 mm height) were correlated with greater MBL (*p* < 0.01 for both variables). Results of multivariable analysis analyzing factors influencing MBL within specific time frames are reported in Table 4.

## 4. Discussion

The present multicenter prospective clinical study showed that platform-switched implants with internal conical connection placed subcrestally presented a reduction in marginal bone levels during the first year of function. Thoroughly analyzing this result, early MBL occurred in the first six months after prosthetic loading. At 15-month follow-up, peri-implant marginal bone levels remained unaltered (difference T4–T3: 0.00 ± 0.19 mm). This finding is in accordance with previous studies on subcrestal implants, showing that MBL mainly occurs in the first period of function [14,21], followed by stabilization of marginal bone levels or even slight marginal bone gain [22].

In the present investigation, a statistically significant positive correlation was demonstrated between the depth of implant insertion and early MBL (*p* < 0.01), confirming a tendency shown in previous studies [22,23]. To be exact, our statistical model suggested that a 1-mm depth increase below the mean implant position at T0 (1.24 mm subcrestal) led to a greater MBL of 0.23 mm at T4. However, from a clinical point of view, it should be underlined that implants with deeper apico-coronal positioning at T0 (>1.5 mm under bone level) resulted in a more subcrestal position at T4, when compared with implants placed more superficially at baseline (<1.5 mm under bone level) (Table 5). Further studies are needed to establish the ideal insertion depth for subcrestal implants, balancing the amount of MBL with the biological shield offered by the presence of bone coronal to the implant shoulder.

General variables, such as age, gender, periodontal status, and smoking habits, appeared not to play a significant role in influencing MBL during the first months of healing. Even if smoking and history of periodontitis are well-known risk factors for the long-term success of implant therapy, their action is time-dependent and often is not predictive for early bone loss after 1-year follow-up [24,25]. Additionally, the one abutment-one time protocol did not have a significant protective action on MBL in comparison to multiple abutment disconnections (three times, in the present study), in agreement with a recent prospective study [26]. This outcome is also consistent with a recent meta-analysis concluding that favorable changes in peri-implant marginal bone level associated with the one abutment-one time protocol should be viewed with caution as its clinical significance remains uncertain [27].

In the present study, the greatest MBL occurred within two months after implant insertion (mean 0.5 ± 0.34 mm), likely due to bone remodeling following surgical trauma and biological width establishment around one-stage implants.

In our sample, variations in implant insertion torque (>50 Ncm vs. ≤50 Ncm) did not influence MBL during the early healing period. This finding is in accordance with some previous clinical trials [28,29] and is in contrast with other studies, showing a negative impact of high torques on marginal bone stability [30,31,32]. However, the relationship between insertion torque and cortical compression, possibly leading to marginal bone resorption, is strictly dependent upon some crucial factors, which were not always adequately controlled in the aforementioned studies: implant crest module design, implant diameter, and cortical bone thickness around implants [6,33,34]. In the present investigation, detrimental distribution of compressive forces to cortical bone following implant insertion was reduced by the subcrestal positioning of implants (not compressing the most coronal part of the cortical bone) and by the crest module design of the fixture used in this study, the platform of which was significantly narrower than the wider part of the implant body (3.3 mm vs. 4.3 mm).

Conversely, all investigated variables involved in biological width establishment (vertical mucosal thickness, healing abutment height, and prosthetic abutment height) had a significant influence on marginal bone remodeling. Biological width is the three-dimensional space necessary for the establishment of a soft tissue barrier around dental implants once they become exposed to the oral cavity [35]. Peri-implant soft tissue can be divided into two main zones: a coronal epithelial portion and a more apical fiber-rich connective tissue [35,36]. Recently, the biological width around two-piece dental implants placed at the crestal level has been measured in human histologic studies. Vertical dimensions varied from 3.26 to 3.6 mm, representing the minimum space required to create an optimal seal and protect the underlying tissue from external agents [37,38]. When vertical space is insufficient for biological width establishment, the healing process includes marginal bone resorption.

In the present study, thin vertical mucosal thickness (≤2 mm), short healing abutments (<3 mm), and short prosthetic abutments (<3 mm) were significantly associated with greater marginal bone resorption (Figure 4). These data are in accordance with numerous clinical trials and a meta-analysis conducted with implants placed at crestal level [17,39,40,41,42,43], even if this matter has been widely debated. In clinical practice and also in the present study, abutment height is adapted to site-specific soft tissue thickness, with the consequence that short abutments have usually been selected in the presence of thin peri-implant mucosa. This condition could represent a confounding factor when analyzing the real influence of both factors on MBL. Recent studies conducted on separate groups (thick mucosa with long and short abutments; thin mucosa with long and short abutments) indicate that MBL during biological width establishment around dental implants placed at crestal level is influenced by abutment height irrespective of vertical mucosal thickness [9,44].

Very few data on these topics are present in literature for subcrestal implants. Some authors suggested adapting the vertical position of implants in relation to soft tissue thickness in order to prevent early MBL [45,46]. The rationale behind this proposal is that subcrestal implant placement could provide additional space for biologic width formation, from the bone crest to the implant platform, resulting in reduced marginal bone remodeling in the presence of thin peri-implant mucosa. However, further clinical and histological studies are necessary to confirm this hypothesis.

Some limitations must be considered when interpreting the outcomes of the present study. These results are not to be generalized for all types of implants: fixtures with a flat-to-flat connection placed subcrestally showed persistent acute inflammation at the microgap between implant and abutment, resulting in increased MBL [15]. The platform-switched conical connection is currently to be considered the pattern of choice to minimize marginal bone remodeling when planning subcrestal implant placement [47].

Another limitation is the use of periapical radiographs to assess marginal bone levels: this method, allowing evaluation of only mesial and distal aspects of peri-implant bone, reduces sensitivity in detecting marginal bone changes.

Moreover, thresholds adopted in the present study to define an abutment as short or long (long: ≥3 mm height; short: <3 mm height) were arbitrary. Future studies should confirm the suitability of these values to define abutment length for implants placed subcrestally.

Finally, the present study collected data from a limited pool of patients in a specific site (posterior mandible): therefore, further trials are needed to generalize these results to a broader population and different areas of the mouth.

## 5. Conclusions

Early marginal bone remodeling around platform-switched implants with conical connection inserted subcrestally was significantly influenced by implant insertion depth and factors related to biological width establishment (vertical mucosal thickness, healing, and prosthetic abutment height). Deep implant insertion, thin peri-implant mucosa, and short abutments were associated with greater marginal bone loss up to six months after prosthetic loading. Peri-implant bone levels tended to stabilize after this time, and no further marginal bone resorption was recorded at twelve months after implant loading.

The outcomes of this study should be confirmed and generalized by further clinical trials with greater numerosity and conducted in different areas of the mouth.

Finally, further investigations are needed to establish the ideal insertion depth for subcrestal implants, balancing the amount of MBL with the biological shield offered by the presence of bone coronal to the implant shoulder.

## Figures and Tables

**Figure 1 jcm-08-01168-f001:**
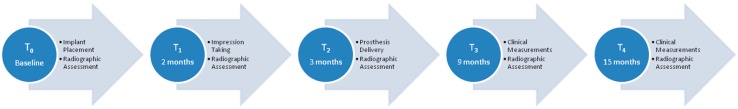
Summary of the visits.

**Figure 2 jcm-08-01168-f002:**
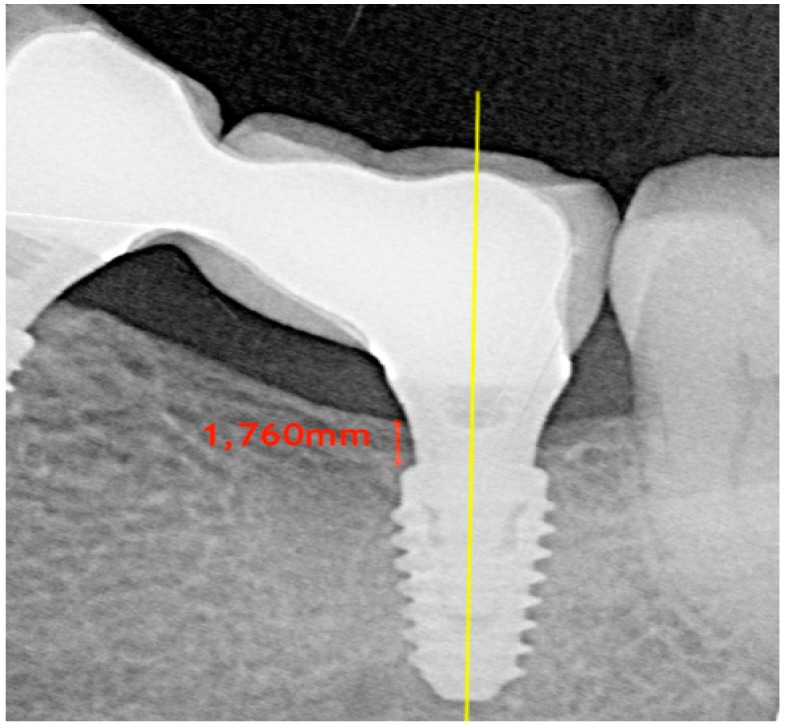
Bone level measurement.

**Figure 3 jcm-08-01168-f003:**
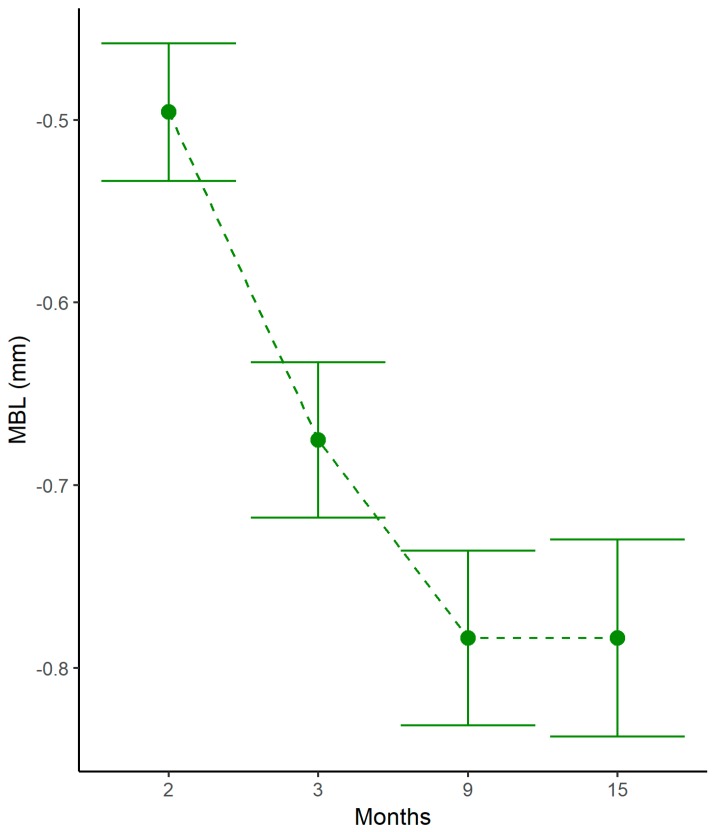
Marginal bone variations at different time points. MBL: marginal bone loss.

**Figure 4 jcm-08-01168-f004:**
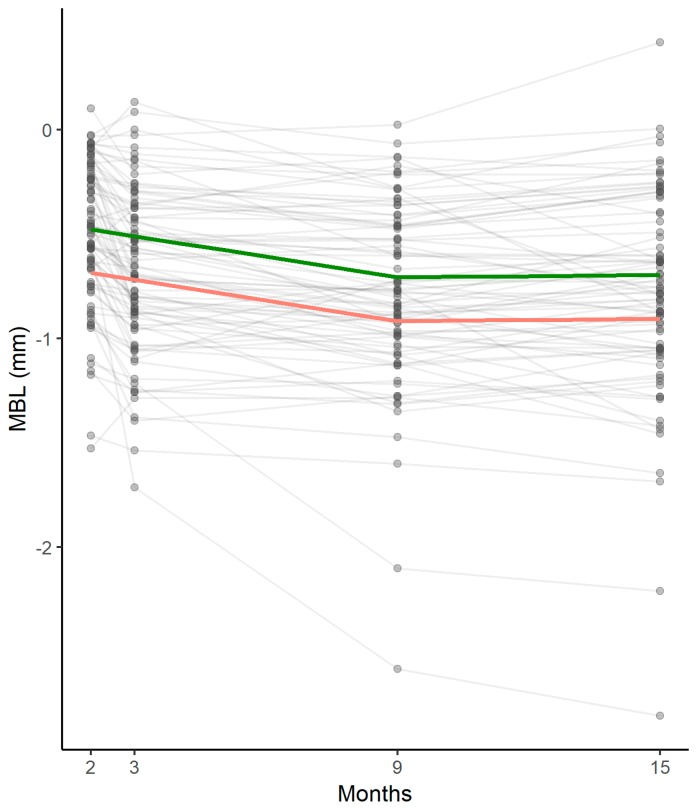
Marginal bone remodeling at different time points estimated by the statistical model: pattern of thick (green) and thin (pink) peri-implant mucosa in the entire sample. MBL: marginal bone loss.

**Table 1 jcm-08-01168-t001:** Characteristics of the included patients.

	Total	%
**Age**		
Male	56.2 ± 13.8	-
Female	59.2 ± 12.3	-
**Gender**		
Male	25	50
Female	25	50
**Smoking Status**		
Smoker	7	14
Non Smoker	43	86
**Periodontal Status**		
Periodontal Health	37	74
Chronic Periodontitis	13	26

**Table 2 jcm-08-01168-t002:** Preliminary univariable linear mixed models, analyzing factors possibly influencing marginal bone loss from implant insertion to 12-months after prosthetic loading.

	MBL (mm)	Std. Error	d.f.	*t*-Value	*p*-Value
**Age**	0.00	0.00	49	−0.49	0.63
**Gender**	−0.02	0.08	49	−0.21	0.83
**Smoking Status**	0.12	0.10	49	1.15	0.26
**Periodontal Status**	−0.02	0.09	49	−0.25	0.80
**Multiple Abutment Disconnections**	0.03	0.08	30	0.43	0.67

MBL: marginal bone loss; Std: standard; DF: degrees of freedom. Adjusted for time effect and depth of implant insertion.

**Table 3 jcm-08-01168-t003:** Multivariable linear mixed models, analyzing factors influencing marginal bone loss from implant insertion to 12-months after prosthetic loading.

	MBL (mm)	Std. Error	d.f.	*t*-Value	*p*-Value	95% CI
**Vertical Mucosal Thickness**	0.24	0.08	30	2.91	0.01	0.07; 0.41
**Depth of Implant Insertion**	−0.23	0.07	30	−3.37	<0.01	−0.37; −0.09

MBL: marginal bone loss; Std: standard; DF: degrees of freedom; CI: confidence interval. Adjusted for time effect.

**Table 4 jcm-08-01168-t004:** Multivariable linear mixed models, analyzing factors influencing marginal bone loss within specific time frames.

	**Time Interval**	**MBL (mm)**	**Std. Error**	**d.f.**	***t*-Value**	***p*-Value**	**95% CI**
**Healing Abutment Height** **Short (vs. Long)**	T0–T2	−0.27	0.07	29	−3.83	<0.01	−0.41; −0.13
**Prosthetic Abutment Height** **Short (vs. Long)**	T2–T4	−0.44	0.07	29	−6.42	<0.01	−0.58; −0.3

MBL: marginal bone loss; Std: standard; DF: degrees of freedom; CI: confidence interval; T0: baseline; T1: 2-month visit; T2: 3-month visit; T4: 15-month visit. Adjusted for time effect, depth of implant insertion, and vertical mucosal thickness.

**Table 5 jcm-08-01168-t005:** Bone loss variations in groups with different implant insertion depth.

Insertion Depth	N° Of Implants	Mean Depth At T0	Mean Depth At T4
>1.5 mm	20	2.01 ± 0.48 mm	0.94 ± 0.76 mm
<1.5 mm	63	1.00 ± 0.34 mm	0.31 ± 0.42 mm

T0: baseline; T4: 15-month visit.

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
