# Peer review of "Factors Influencing Early Marginal Bone Loss around Dental Implants Positioned Subcrestally: A Multicenter Prospective Clinical Study"

_jcm, 2019, doi:10.3390/jcm8081168_

Round 1

Reviewer 1 Report

I believe the manuscript needs to be revised to improve English language and style of the paper.

The description of Table 1 is vague and needs to be changed.

It is better to state implants placed sub-crestal instead of using implants positioned under bone level.

Author Response

First of all we want to apologize because the Editorial Office, during files conversion of the first submitted version of the manuscript, likely lost Figure 1 and some parts of the figure and table captions. The result was that the illustrated part of the manuscript was a bit confused. In this revised version we hope everything will work properly.

Thanks for your suggestions for improving the manuscript.

This is point-by-point response. All the modifications in the manuscript are highlighted in yellow.

I believe the manuscript needs to be revised to improve English language and style of the paper.

A professional translator entirely revised the manuscript

The description of Table 1 is vague and needs to be changed.

Table descriptions were modified

It is better to state implants placed sub-crestal instead of using implants positioned under bone level.

The title of the paper was modified according to your suggestions

Reviewer 2 Report

Overall;

This paper assesses how long after the implant treatment the initial bone changes are likely to occur, and discusses the factors that are expected to affect the initial bone changes.

However, the statistical methods used and their results are very difficult to understand, and it is unclear what the numbers shown in the table indicate.(For example, "Value" in Tables 2 to 4.)

In addition, the methods of selecting the subjects are described in the result. These should be mentioned in the Material and Methods chapter.

Individual indications;

-Figure 3 was not found unless I was wrong...

-Figures 1 and 2 are integrated, but this figure probably only shows the change of the average value of MBL. So, readers cannot understand at all the explanation of the implant used in this study and the measurement method of the radiograph. Readers may not all be dental professionals.Therefore, you should explain what kind of dental implant was used and how to measure the radiographs using the figure suitably.

-It is recommended that the contents of section 3.1 of the results chapter, including Table 1, be described in the Material & Methods chapter.This change makes the subject clearly expressed.

-It is difficult to understand the meaning of the numerical values shown in Tables 2 to 4.The meaning of these values should be indicated in the text or in the description of the table.

Author Response

First of all we want to apologize because the Editorial Office, during files conversion of the first submitted version of the manuscript, likely lost Figure 1 and some parts of the figure and table captions. The result was that the illustrated part of the manuscript was a bit confused. In this revised version we hope everything will work properly.

Thanks for your suggestions for improving the manuscript.

This is point-by-point response. All the modifications in the manuscript are highlighted in yellow.

Individual indications;

-Figure 3 was not found unless I was wrong...

As we said, we tried to add again properly all figures and tables

-Figures 1 and 2 are integrated, but this figure probably only shows the change of the average value of MBL. So, readers cannot understand at all the explanation of the implant used in this study and the measurement method of the radiograph. Readers may not all be dental professionals.Therefore, you should explain what kind of dental implant was used and how to measure the radiographs using the figure suitably.

Figures were re-arranged. I hope now it is more understandable.

-It is recommended that the contents of section 3.1 of the results chapter, including Table 1, be described in the Material & Methods chapter.This change makes the subject clearly expressed.

Respectfully, we think that the contents of section 3.1, even if they are undoubtedly helpful for clarification, should remain in Results section, as they report the key findings of the study AFTER therapy and measurements were performed.

-It is difficult to understand the meaning of the numerical values shown in Tables 2 to 4.The meaning of these values should be indicated in the text or in the description of the table.

The table was modified as suggested

Round 2

Reviewer 2 Report

It was improved that the figure or the table that was not displayed correctly because of the trouble in submitting. However, it is easier to understand how to measure intra-oral X-ray images using the example image. Please add it.

Author Response

t was improved that the figure or the table that was not displayed correctly because of the trouble in submitting. However, it is easier to understand how to measure intra-oral X-ray images using the example image. Please add it.

Thanks for your suggestion. As requested, we added Figure 2 to clarify with an example the measurement method of the present study.